# Societal and organisational influences on implementation of mental health peer support work in low-income and high-income settings: a qualitative focus group study

Mary Ramesh ,[1] Ashleigh Charles ,[2] Alina Grayzman ,[3] Ramona Hiltensperger ,[4] Jasmine Kalha ,[5] Arti Kulkarni ,[5] Candelaria Mahlke ,[6] Galia S Moran ,[7] Richard Mpango ,[8,9] Annabel S. Mueller-Stierlin ,[4] Rebecca Nixdorf ,[10] Grace Kathryn Ryan ,[11] Donat Shamba ,[1] Mike Slade [2,12]

MR and AC contributed equally. DS and MS contributed equally.

MR and AC are joint first authors. DS and MS are joint senior authors.

For numbered affiliations see end of article.

**Correspondence to**
Donat Shamba;
dshamba@ihi.or.tz

## ABSTRACT

**Objectives** Despite the established evidence base for mental health peer support work, widespread implementation remains a challenge. This study aimed to explore societal and organisational influences on the implementation of peer support work in low-income and high-income settings.

**Design** Study sites conducted two focus groups in local languages at each site, using a topic guide based on a conceptual framework describing eight peer support worker (PSW) principles and five implementation issues. Transcripts were translated into English and an inductive thematic analysis was conducted to characterise implementation influences.

**Setting** The study took place in two tertiary and three secondary mental healthcare sites as part of the Using Peer Support in Developing Empowering Mental Health Services (UPSIDES) study, comprising three high-income sites (Hamburg and Ulm, Germany; Be'er Sheva, Israel) and two low-income sites (Dar es Salaam, Tanzania; Kampala, Uganda) chosen for diversity both in region and in experience of peer support work.

**Participants** 12 focus groups were conducted (including a total of 86 participants), across sites in Ulm (n=2), Hamburg (n=2), Dar es Salaam (n=2), Be'er Sheva (n=2) and Kampala (n=4). Three individual interviews were also done in Kampala. All participants met the inclusion criteria: aged over 18 years; actual or potential PSW or mental health clinician or hospital/community manager or regional/national policy-maker; and able to give informed consent.

**Results** Six themes relating to implementation influences were identified: community and staff attitudes, resource availability, organisational culture, role definition, training and support and peer support network.

**Conclusions** This is the first multicountry study to explore societal attitudes and organisational culture influences on the implementation of peer support. Addressing community-level discrimination and developing a recovery orientation in mental health systems can contribute

## STRENGTHS AND LIMITATIONS OF THIS STUDY

⇒ The sample size (86 participants, across 12 focus groups) and sampling across two low-income and three high-income sites increases the credibility of the findings and their relevance to similar settings.

⇒ Independent coding by multiple analysts from different cultures enhances trustworthiness.

⇒ Sociodemographic characteristics were not sufficiently collected to be reported, limiting transferability of findings.

⇒ Study participants were peer support workers and mental health professionals; there is a need to conduct further studies with service users or recipients of peer support services in order to understand their perceptions on the influences on the implementation of peer support work.

⇒ Two focus group discussions per site may not reach saturation; however, the study involved different sets of respondents to bring together the perspectives of different groups who either had experience in peer support work or were planning to use peer support workers.

to effective implementation of peer support work. The relationship between societal stigma about mental health and resource allocation decisions warrants future investigation.

**Trial registration number** ISRCTN26008944.

## INTRODUCTION

Many people living with severe mental illness do not receive adequate care. For example, in Europe, the EuroPoPP-MH study found that a comprehensive range of community-based services existed in only 8 of 29 countries.[1] The resulting gap between demand and supply is called the treatment gap or care gap.[2] While

mental health has been identified as a global priority,[3] the mental health treatment gap remains and is largest in low-resource settings.[4]

One reason for the treatment gap in low-resource settings is that mental health initiatives do not always sufficiently address contextual aspects, such as upstream social determinants, geographical and linguistic differences, and sociodemographic influences such as ethnicity, caste and tribe.[5] This leads to barriers in receiving mental health treatment, including stigma, social exclusion and inequities in terms of resourcing.

Mental health peer support is an established intervention involving a person with lived experience of mental health problems and recovery employed to offer support to others with mental health problems. Peer support workers (PSWs) act as credible role models of recovery,[6] instilling hope through positive self-disclosure, modelling the use of experiential knowledge for self-care and offering supportive relationships based on intentional mutuality.[7] There is a growing empirical evidence base for PSWs.[8–11] A recent systematic review identified 19 randomised controlled trials,[12] all from high-income countries. This review found PSW was associated with beneficial outcomes in relation to supporting recovery, empowerment and social networks. However, heterogeneity in the implementation of peer support was identified as an important knowledge gap.

Most research on mental health peer support work has been conducted in high-resource (typically Anglophone) settings, including creation of core PSW principles[13 14] and evaluation.[8] However, PSW roles are increasingly being developed, formalised and implemented in more diverse settings, such as China,[15] India,[16] Israel,[17] Singapore[18] and Uganda.[19] An important knowledge gap, therefore, exists in relation to PSW implementation influences across settings with different resource levels.[20] A recent systematic review synthesised 53 studies (none from low-income countries) to identify 14 influences on implementation of mental health PSW.[21] The most commonly reported influence was organisational culture, identified in 53% of studies, followed by training and role definition. Societal influences were also identified, including PSW access to a peer network, resource availability and financial arrangements.

The Grand Challenges for Mental Health initiative identified the importance of research along the translation continuum including implementation, and emphasised that implementation is a challenge not just in low-income and middle-income countries.[3] In other words, the focus should be on implementation research including both lower-resource and higher-resource settings. To our knowledge, no study has explored PSW implementation across multiple countries. The aim of this study was to explore and characterise the societal and organisational influences on the implementation of mental health peer support work in low-income and high-income settings.

## METHODS

This study was conducted as part of UPSIDES (Using Peer Support in Developing Empowering Mental Health Services), a 5-year (2018–2022) European Union-funded multinational study that aims to replicate and scale up peer support interventions for people with severe mental illness.[22]

### Design

A qualitative research design informed by a critical realist perspective was used. A critical realist approach was chosen as it can help in identifying some of the underlying organisational and societal influences of PSW implementation. Focus groups were selected over other data collection approaches to maximise breadth of data coverage.

### Setting

Data were collected from five UPSIDES study sites. Sites were based in two high-income countries (Hamburg and Ulm sites, Germany; Be'er Sheva, Israel) and two low-income countries (Dar es Salaam, Tanzania (low-resource setting at the time of data collection, rebanded in 2020 to lower-middle income country); Kampala, Uganda), ensuring regional diversity (Europe, Eastern Mediterranean, sub-Saharan Africa). Sites were classified as low-resource settings because they are based in low-resource countries. As previously reported,[21] sites were also diverse in terms of their experience with peer support work, with two sites (Dar es Salaam, Ulm) having no or very little previous experience.

### Participants

Participants were purposively selected to include stakeholders with different perspectives on PSW implementation: actual or potential PSWs, mental health clinicians or managers from hospitals or community services; and regional or national policy-makers. To be included, participants had to be over 18 years of age and capable of providing informed consent.

Participants came from a range of community-based, outpatient and inpatient mental health services in Germany, Uganda and Tanzania, and from a range of community mental health rehabilitation services in Israel. In all sites, multidisciplinary inpatient and outpatient care involves psychotherapy, psychosocial rehabilitation and psychiatric clinics, with some sites also offering family intervention, vocational skills training, cognitive enhancement therapy, psychoeducation, predischarge social interventions and physical healthcare.

### Procedures

A conceptual framework—a network of interlinked concepts together providing a comprehensive understanding of a phenomenon[23]—was developed to capture the key elements and implementation influences on the PSW role. The conceptual framework comprised (A) PSW principles and (B) societal and organisational implementation influences, as shown in table 1. The PSW principles

**Table 1** Conceptual framework for PSW principles (n=8) and societal/organisational implementation influences (n=5)

| Principle | Definition |
|---|---|
| 1. Mutual | PSWs have similar experiences to peer support users |
| 2. Reciprocal | PSWs and peer support users both give and receive in the relationship |
| 3. Non-directive | PSWs develop solutions together with the peer support user, instead of dictating solutions |
| 4. Recovery focused | PSWs support the peer support user on his/her path towards overcoming the problems that they experience |
| 5. Strengths-based | PSWs show a positive attitude and identify and build on the strengths and recovery progress of peer support users |
| 6. Inclusive | PSWs do not exclude people on the basis of the nature of their problems or beliefs about their level of ability, and help peer support users to find their place in society |
| 7. Progressive | PSWs and users advance together towards recovery, this is not a befriending relationship that aims to maintain current progress |
| 8. Safe | PSWs and users develop a common basis of trust and safety, which is central to the planning of the service and training of peer workers |
| Implementation influence | Description of societal/organisational influence |
| 1. Group versus individual | Peer support can be offered in single sessions and in a group setting |
| 2. Extent to which both parties choose to enter the relationship | Peer support pairs and groups can be formed by the organisation, but also by the peers themselves |
| 3. Extent to which rules govern the relationship | There can be implicit and explicit rules underpinning how the peer support work is conducted |
| 4. Extent to which the parties involved are in the same place in their recovery journey | Depending on the state of recovery, peer support users can become PSWs and vice versa |
| 5. Extent to which the PSWs focus on peer support users | PSWs can support recovery for peer support users and/or promote a recovery orientation for the staff they work with, the institution they work in and the society they live in |

PSW, peer support worker.

were derived from a researcher-led integration of established core principles from high-resource settings.[13 14 24] At the time of development (2017), there was an absence of integrated evidence, so a systematic review was subsequently undertaken,[21] but for the current study the implementation influences were developed through consultation with experts in the UPSIDES consortium.

The conceptual framework informed the development of a topic guide (online supplemental file 1), comprising open-ended conversational prompts to explore the cultural applicability of PSW principles and to identify societal and organisational implementation influences. Exploration of areas of disagreement was encouraged, as was speculation about potential implementation influences in sites with no experience of PSW. The topic guide was developed in English, commented on by all sites, and then finalised and translated into Kiswahili (Dar es Salaam), German (Hamburg/Ulm), Hebrew (Be'er Sheva) and Luganda (Kampala).

Focus groups were conducted at each site between September and December 2018. In each site, potential participants were identified by mental health clinicians and UPSIDES research workers. Two focus groups were conducted per site, apart from Kampala where four focus groups (two for PSWs, two for other stakeholders) and three individual interviews were conducted. All focus groups were conducted in the local language and held in a health service or community venue.

Each focus group comprised five to nine participants, and lasted up to 60 min. Facilitators were UPSIDES research workers from the site, who were bilingual in the local language and English, and came from psychology, sociology, health sciences, social work and nursing backgrounds. All facilitators were experienced in qualitative data collection, and actively managed group dynamics to ensure full participation from all participants. Focus groups were recorded using an audio recorder and researchers took field notes during the discussions. After the focus groups, local language transcripts were made, with pseudonymisation of identifying information about participants and third parties. Each local language transcript was translated into English by the local UPSIDES researcher, and checked by the UPSIDES translation leads (Nottingham, UK and Pune, India) for data integrity, identifying points for site checking if needed. Finalised transcripts were password protected and uploaded to a restricted area on the UPSIDES website.

### Analysis

A combination of deductive and inductive thematic analysis was conducted.[25] The two primary coders were UPSIDES research workers in Dar es Salaam (MR: background

in public health) and Nottingham (AC: mental health nursing, sociology). MR and AC independently read all transcripts to familiarise themselves with the content and start the process of creating preliminary codes and categories. Coding was then discussed with site leads in Dar es Salaam (DS: social science) and Nottingham (MS: clinical psychology), following which a preliminary coding framework for implementation influences was developed. The codebook was then transferred into NVivo V.12 software for coding. MR and AC independently coded the same four transcripts, and then discussed and reviewed any differences or discrepancies and any additional themes that emerged from the data. Following review, refinement and defining of themes, an agreement was reached, and new codes were incorporated into the final coding framework. The remaining transcripts were then coded with repeated discussion between coders. The finalised coding framework was iteratively discussed among the four primary analysts (AC, MR, MS and DS) and the wider author team until a consensus was reached.

## Patient and public involvement

Individuals with lived experience are involved at multiple levels of the UPSIDES Study, including as part of the site team, as advisory board members, as PSWs and as authors on some papers. No further patient and public involvement specific to the current study was undertaken.

## RESULTS

A total of 86 individuals participated across 12 focus groups. These include focus groups in Ulm (n=2), Hamburg (n=2), Dar es Salaam (n=2), Be'er Sheva (n=2) and Kampala (n=4, two for PSWs, two for other stakeholders). In addition to the focus groups, Kampala also conducted three individual interviews. Details are shown in table 2.

Six implementation influence themes were identified: community and staff attitudes, resource availability, organisational culture, role definition, training and support and peer support network.

## Theme 1: community and staff attitudes

Community and staff attitudes towards mental illness were perceived to be both a barrier and also a facilitator for PSWs to perform their roles. Some participants, especially in lower-income countries, reported that people with mental health conditions are considered inferior and are also rejected, thus making it difficult for the PSWs to perform their roles effectively.

> When it comes to class, mental health patients are considered second-hand, third-hand or fourth-hand citizens. So we are marginalised among the marginalised. We take the lowest rank status point in the community. [#9, Kampala, PSW]

> Even in our community a person with a mental illness is not a priority. A large percent of our patients live in a community where there is stigma to the extent that they are not brought to the hospital. [#2, Dar es Salaam, Mental Health Clinician]

Furthermore, religious beliefs can also act as a barrier in implementing peer support work, as was apparent mainly in the lower-income countries.

> There are so many religious leaders who believe that God doesn't fail. They interfere with our work. They stop our patients from taking medicine and they say that God is going to perform miracles then in the end they relapse. The traditional healers believe that mental illness is caused by traditional issues and they don't need Western medicine, they need herbs. [#26, Kampala, PSW]

Also in the lower-income countries, PSWs reported experiencing rejection when they go to visit service users, as some family members do not want the mental health status of their relative to be revealed, thus making it difficult for the PSWs to perform their roles as described by a PSW from Uganda.

> We are rejected, you can go to that person's place who may not wish to see you and they don't welcome you and you can't insist. Sometimes they just avoid you. [#12, Kampala, PSW]

**Table 2** Focus group participants (n=86)

| Site | Focus groups | n | Participant | | | | Gender | |
|---|---|---|---|---|---|---|---|---|
| | | | Potential or actual PSW | Mental health worker | Mental health manager | Policy-maker | Male | Female |
| Ulm | 2 | 12 | 1 | 10 | 0 | 1 | 4 | 8 |
| Hamburg | 2 | 12 | 7 | 5 | 0 | 0 | 4 | 8 |
| Kampala | 4 | 32 | 16 | 14 | 1 | 1 | 10 | 22 |
| Dar es Salaam | 2 | 16 | 0 | 12 | 4 | 0 | 7 | 9 |
| Be'er Sheva | 2 | 14 | 2 | 8 | 4 | 0 | 5 | 9 |
| Total | 12 | 86 | 26 | 49 | 9 | 2 | 30 | 56 |

PSW, peer support worker.

For participants in lower-income countries, community initiatives through arts, media and local projects raised awareness in the community and educated people about mental health. These initiatives also enabled the PSWs to be known as role models in the community and have inspired hope to others. Additionally, the notion of knowledge from experience adds value to the potential contribution of the PSW and helps transform and enhance the value of lived experience.

Now that we are role-models in the community people inquire from us about the things which I did to enable me to stabilize while at first they were stigmatizing me. They were beating me but now it is in their families and they are having issues worse than mine. They are like, 'You see that mentally sick lady who was here? She is now stable. Let us go and inquire from her so that she can help us'…So we have become brokers in the village. [#14, Kampala, PSW]

In both the lower-income and higher-income countries, PSWs reported facing stigma from health service providers. PSWs are labelled as 'mad', and in some cases, health workers question why a person with a mental illness is part of the staff team, as described by PSWs from Israel and Uganda, below.

Stigma prevails mainly among doctors and employees in medical and rehabilitation services. I blame it on the illness model as perceived by most. The model holds that illness is an inherent state, a permanent life solution. In my opinion, this is the core of the problem. Even if you have stopped medication and have been well for ten years, still the label remains. It sounds like it's for life. We should stop labelling. [#4, Be'er Sheva, PSW]

And even here at [name] hospital some of the professionals say, 'Who can work with those mad ones? But some of them, those who accepted us, are happy to work with us. They even smile at us and talk to us, but there are others who think that mental illness is contagious. [#5, Kampala, PSW]

Acceptance of the PSWs by the health service providers plays a significant role in facilitating the implementation of peer support services. PSWs' lack of acceptance from health service providers and unwillingness to work with PSWs can cause them to become 'unstable' and fail to fulfil their duties, as described here by a participant from a higher-income setting.

Not being accepted made the PSW to be alone. You find that a PSW is stable in the beginning but you notice that she was destabilized in the course of being a PSW due to the pressure from outside and lack of acceptance from the team. [#009, ULM, Mental health clinician]

Although there are some health providers who stigmatise PSWs, there are others who think that PSWs are an asset to both health service providers and the recipients of peer support services. One participant from a lower-income country perceived that PSWs act as a bridge between the mental health workers and the service users. PSWs and their peers share a mutual relationship, and peers open up more to the PSWs than to mental health staff.

Actually it has bridged a gap between service users and service providers. There is some kind of mutual understanding that we have built up. We are treated like staff. [#14, Kampala, PSW]

## Theme 2: resource availability

Providing resources for PSWs to carry out their work is an important factor influencing the provision of peer support services. Several participants reported that PSWs have limited resources in terms of money for airtime (using their phone for work-based calls), transport to visit individuals in the community, and payment to cater for their daily needs when performing their role, especially in lower-income settings. Two PSWs described the financial challenges they face when working in the community.

PSWs should receive financial support so as to be able to make home visits to service users. They should also be incentivized so as to deal with the different challenges that they face in the community. [ #1, Dar es Salaam, Mental Health Clinician]

According to the inflation in the country, the money can't be enough to move [travel] to a community. Sometimes you need to buy airtime to call a peer, you have to fix something or food in the community, you have to get something to drink or eat in the community. Sometimes you find that this peer you are visiting is far away from where you stay, so the money we are paid is not enough. [#10, Kampala, PSW]

Participants from higher-income countries also reported that, while PSWs are an important component of mental health services, and while this is reflected in policy, there is a limited budget set for them, and there is a particular challenge in relation to funding arrangements for the peer support programme. Differing funding arrangements across organisations and systems means that the expectation for PSWs to be employed in many departments is a challenge due to limited resources which are often stretched to cover a range of competing and differing organisational needs. In addition, uncertainties around who funds PSW programmes means that cross-funding from other projects is common.

The current policy needs to change. The policy says peer specialists have to be everywhere…and it requires resources which we don't have. I am not sure if I can raise the issue, but we don't have budgets like the welfare…We need to get a special budget for the program. [#6, Be'er Sheva, Mental health clinician]

Then there is always the question of who finances it. For example, the peer support workers on the ground floor (acute ward), are they financed by the ward budget or hospital budget or are they somehow cross-financed by other projects? [#3, Hamburg, Mental health clinician]

The facilities available to enable PSWs to perform their roles can be inadequate. It was noted that the working environment and infrastructure in lower-resource health facilities is very poor, exposing the PSWs, mental health workers and services users to many risks. Two participants described in detail the workplace environment and the impact of this for individuals using the services and for workers.

The…Outpatient department can only accommodate four people while there are almost a hundred or ninety people per day, so you find that people are just standing. [#1, Dar es Salaam, Mental health clinician]

We don't have enough facilities within the hospital, nurses face some challenges also. When you go to [name] ward, some peers are sleeping down [on the floor] and even divide blankets. You may find that one blanket is divided among 2 to 4 patients. [#13, Kampala, PSW]

### Theme 3: organisational culture

The goals, attitudes, role assumptions and values held by the organisation about PSWs, and the relationship between the PSW and the organisation, are important for PSW implementation. Participants mainly in higher-income countries reported that working inside structured and hierarchal systems can create a feeling of indebtedness to the organisation which can impact on PSWs' autonomy in decision-making and contribute towards feelings of disempowerment.

When you enter a job as a consume-provider, at least in the beginning, there is part of you that feels like they are doing you a favour that they hired you. That you have to do what the organisation tells you to do in order to gain experience, etc. [#8, Be'er Sheva, PSW]

For successful implementation of peer support work, participants explained how PSWs should work by following the rules that have been set up in the organisation, and the organisation should also adjust its system to accommodate the PSW. For example, organisational flexibility and understanding of the role was perceived as crucial. This included accommodations in the workplace which allow PSWs to manage their own mental health and carry out their role effectively. However, participants acknowledged that organisations' expectations were important in terms of PSWs being recognised as members of the team and as part of the organisation. Participants highlighted that organisational rules, processes and structures were not always easy to manage or negotiate for PSWs.

The PSW needs to understand that he is coming here as a worker and needs to follow the expectations like any other worker. As a worker he is also entitled to some sort of accommodation system, for example it might be difficult for them to start work in the mornings because they have to take pills, so he will start at 10:00 and not 7:30. So he'll do more afternoon shifts as opposed to morning ones. [#2, Be'er Sheva, Mental Health Clinician]

Participants highlighted that the formalisation of the PSW role in such defined systems raises the question of how much of the role should remain informal vs formal in order for PSWs to fit in. For example, the integration of PSWs into teams that already have clearly defined roles, responsibilities and hierarchies raised uncertainties around what this might mean for the PSW role and how the introduction of the PSWs into formal systems may impact on role integrity.

I also think that what especially happens with peer specialists, is some sort of formalization of this thing, and how much do we really actually want to formalize it. And how much of it do we have to keep informal, which is one of the worries or dilemmas. The ideals that are really inside of this system that is so formal, and is hierarchical and clear. [#3, Be'er Sheva, PSW]

In lower-income countries, participants reported that the support can be very limited, due to the lack of psychological and social support resources available. Participants recognised the importance of these resources for carrying out their role, but also acknowledged the difficulties and ongoing challenges of working within a system with limited resources and that follows a strongly medical model. However, in these settings, attempts were made to provide this support as much as possible despite these challenges, as one participant describes.

In summary, there is diagnosis, treatment mostly pharmacological using medical treatment. [We] do our best to try to provide psychological and social support but those are very limited most of the time. [#30, Kampala, Manager]

### Theme 4: role definition

Nearly all participants reported that having a clear role definition and expectations was important, because without this reference point, potential role confusion and uncertainty ensue.

I don't think it's that easy. They often don't know what they can do themselves. That they also have ideas, what can I actually do now? And I don't think there were enough guidelines or terms of reference. [#11, Ulm, Mental health clinician]

In addition, most participants identified that the wide variety of tasks PSWs can perform means it can be difficult to construct a role description that accurately fits

with real-life peer support practice. Participants spoke in detail about how the role is performed differently depending on where the PSW works, and many participants highlighted how individual differences were also considered for specific PSW roles.

> The task fields of peer support workers are totally different. That is always person-dependent. We tried to create a kind of job description already and that was very, very difficult. Because we didn't want to restrict the peer support workers too much. Since the tasks always depend on the personality. [#2, Ulm, Mental health clinician]

### Theme 5: training and support
Training for PSWs and health workers is an important factor for successful implementation of PSW roles. Some participants in both lower-income and higher-income countries described peer support services as something which is new to other mental health professionals, meaning that some lack knowledge of what peer support work is and what PSWs can do. Training healthcare workers and PSWs will help in reducing uncertainties among professionals.

> Peer support work is something new in other countries, it needs to be introduced to other staff members in a larger scale. There is also a simple lack of knowledge, not just supervision, but knowledge of what it is all about, and this lack of knowledge also leads to uncertainty among professionals. What, how do we deal with it now, what do we trust them to do, what do we take away from them because it is not the right thing to do? [#11, ULM, Mental health clinician]

The availability of initial training enables PSWs to know what is expected from them and also understand their needs. Initial training was identified as key for PSWs being prepared for working in the role from both lower-income and higher-income countries. Many participants highlighted that training which provides an understanding of the different types of PSW activities and the work-based challenges PSWs may face was important. Some examples of the training content identified as key for initial training included knowledge of the varying attitudes towards mental health, working with individuals in distress, their families, PSWs and other mental health workers.

> The first thing that they are supposed to receive is training. If they receive training they will know their job description and the techniques of going to the families because there are families which don't want people to know that they have a mental patient. [#3, Dar es Salaam, Mental health clinician]

> I can say that the PSW program that I was part of had PSWs who first of all received training especially to understand their needs, making sure they are dealing with mental illness of others and also how they work with PSWs. [#30, Kampala, Manager]

Most participants also identified the need for further and ongoing training opportunities to be provided. Ongoing training that was highlighted as important for PSWs included understanding boundaries, knowledge about the code of conduct and levels of disclosure. Continual training was viewed as an expectation that should be in place and carried out, so PSWs can continue to carry out their role effectively along with developing knowledge and learning new skills.

> Peer support workers need more training, continuous training. Even if the training is a one-off. So this should be happening. It shouldn't be a big deal. [#30, Kampala, Manager]

Participants in both lower-income and higher-income countries described how PSWs need support to maintain their well-being and carry out their roles. Guidance, supervision and support from other healthcare workers is very important for PSWs in carrying out their work, from a practical perspective. It also enables them to feel part of the wider team, rather than feeling as though they are working in isolation. Participants from both high-income and lower-income countries described the importance of ongoing support.

> Coming from a person who is currently dealing with very active symptoms with varying levels of force, a person needs…regulatory capacity, the ability to manage workloads, the ability to receive help, to be helped and to defend oneself. [#1, Be'er Sheva, Mental health clinician]

> They shouldn't always work in isolation; they should be supervised. [#24, Kampala, Mental health clinician]

> Peer support workers can't be independent, they need professional community nurses to guide them so they can go out in the field, they can be together. [#30, Kampala, Manager]

### Theme 6: peer support network
Participants in both lower-income and higher-income countries explained that having access to a peer network enables PSWs to address their potential challenges, makes them feel connected through sharing their experiences and also enables them to feel stronger together. Further, through these networks PSWs can get to know each other and identify if another PSW is facing a problem, looking after the well-being of the network's members.

> I can see how a group of peers impact each other non-stop and advances processes almost as if it is a race, but not in a bad way. That is to say, not from a place where you feel that they are forcing you to run, but from a place where a lot of people who are together all the time, are shattering stigmas about one another, I think that a group is stronger. [#3, Be'er Sheva, PSW]

We peers have what they call buddies. A buddy is a person who knows more about you whereby in case you show signs of relapse that buddy will say, '(Name) is getting a relapse, (name) do this and this'. He will help you and bring you medical personnel and overcome the situation. [#11, Kampala, PSW]

## DISCUSSION

Our study identified six influences on PSW implementation. At the societal level, community stigma and lay beliefs about mental health conditions were influential. At the organisational level, the interlinked themes of resource allocation and organisational culture were identified, as well as staff attitudes and the challenge of ensuring role clarity. At the PSW level, both adequate training/support and a strong peer support network were facilitators of implementation.

Two aspects of our findings are noteworthy in relation to other studies. First, the conceptual framework was developed on the basis of research almost exclusively from high-income countries, and identified PSW and organisational influences. The implementation influences identified by participants had a stronger emphasis on societal aspects, including attitudes and role assumptions. Our findings are consistent with the previously discussed systematic review,[21] published since the conceptual framework was developed. This validates the importance of considering organisational and societal aspects when implementing PSW in different resource settings.

This involves developing community awareness regarding the value of peer support, to gain the support of family and community members.[26] Second, the PSW-level influences indicate the need to modify how PSW is provided in different settings. A systematic review of 39 studies (only 1 from a lower-income setting), identified 7 types of modification to the PSW role,[27] including recruitment processes, role expectations, training and support. Recent research is expanding to also consider staff attitudes,[28] organisational integration of PSWs,[29 30] organisational climate[31] and context.[32 33]

The primary implication is that more attention needs to be paid to societal attitudes and organisational culture in developing and implementing PSW programmes. Discrimination and stigma relating to mental health are global challenges,[4] but our findings suggest that there is a relationship between community attitudes and the ability to involve people with lived experience in the mental health workforce as PSWs. In terms of organisational culture, the findings reinforce existing evidence[34] that organisational culture impacts on recovery support, so organisational transformation may be needed.[35] Approaches to supporting culture change within mental health services include the introduction of prorecovery interventions,[36 37] development of adjunctive services such as Recovery Colleges,[38] working with teams[39] and introducing coproduction[40] and growth-oriented approaches.[41]

A better understanding of the relationship between the identified influences is needed. In UPSIDES, the theory of change technique is being used to map out different steps in the implementation of the PSW intervention, and to articulate the connections between these steps. The impact of societal and organisational influences on PSW effectiveness will be further explored in the multinational UPSIDES randomised controlled trial (ISRCTN26008944) which is currently underway.[42]

The strengths of the study include the sample size, the use of multiple informants, using local language topic guides to avoid excluding non-English speaking participants and the multinational sample. Credibility of the findings was enhanced by independent coding and the use of multiple analysts.

Several limitations can be identified. One significant shortcoming is that sociodemographic characteristics of participants were not collected in a standardised way across all sites, so are not reported here—limiting the transferability of findings. While the sample is large for a qualitative study, the findings are complex and nuanced, so our analysis focused on semantic rather than latent coding.[25] Future analysis might explore the relationship between the identified implementation influences, such as how community attitudes may distally impact on resource allocation. While the use of analysts with different professional backgrounds reduced researcher influence on findings, the credibility of the findings could be enhanced by member checking, and including people with lived experience as coanalysts.[43] Finally, the relatively small number of policy-maker participants may account for the limited mention of national and regional policy as an influence.

## CONCLUSIONS

This is the first multicountry study to explore societal attitudes and organisational culture influences on the implementation of peer support. Addressing community-level stigma and discrimination and developing a recovery orientation in mental health systems can contribute to effective implementation of peer support work.

**Author affiliations**
¹Department of Health System, Impact Evaluation and Policy, Ifakara Health Institute, Dar es Salaam, Tanzania
²School of Health Sciences, University of Nottingham, Nottingham, UK
³Department of Social Work, Ben-Gurion University of the Negev Faculty of Health Sciences, Beer Sheva, Southern, Israel
⁴Department of Psychiatry II, Ulm University, Ulm, Germany
⁵Centre for Mental Health Law and Policy, Indian Law Society, Pune, India
⁶Department of Psychiatry, University Medical Center, University Medical Center Hamburg-Eppendorf, Hamburg, Germany
⁷Department of Social Work, Ben-Gurion University of the Negev, Southern Israel, Israel
⁸Butabika National Referral Hospital, Kampala, Uganda
⁹Department of Mental Health, School of Health Sciences, Soroti University, Soroti, Uganda
¹⁰Department of Psychiatry, University Medical Centre Hamburg-Eppendorf, Hamburg, Germany

[11]Centre for Global Mental Health, London School of Hygiene & Tropical Medicine, London, UK

[12]Faculty of Nursing and Health Sciences, Health and Community Participation Division, Nord University, Postbox 474, 7801 Namsos, Norway

**Acknowledgements** The Using Peer Support In Developing Empowering Mental Health Services (UPSIDES) Study is a multicentre collaboration between the Department for Psychiatry and Psychotherapy II at Ulm University, Germany (Bernd Puschner, coordinator); the Institute of Mental Health at University of Nottingham, UK (Mike Slade); the Department of Psychiatry at University Hospital Hamburg-Eppendorf, Germany (Candelaria Mahlke); Butabika National Referral Hospital, Uganda (Juliet Nakku); the Centre for Global Mental Health at London School of Hygiene & Tropical Medicine, UK (Grace Ryan); the Department of Health Systems, Impact Evaluation and Policy, Ifakara Health Institute, Dar es Salaam, Tanzania (Donat Shamba); theDepartment of Social Work, Ben-Gurion University of the Negev, Southern Israel (Galia Moran) and the Centre for Mental Health Law and Policy, Pune, India (Jasmine Kalha).

**Contributors** MR and AC contributed to data acquisition, analysis and interpretation and drafting the manuscript. AG, RH, RN, AK, JK, CM, GSM, RM, AMS, GKR, DS and MS contributed to the design of the work, data acquisition, analysis and interpretation, and critically revised the work for important intellectual content. MR, AC, AG, RH, RN, AK, JK, CM, GSM, RM, AMS, GKR, DS and MS gave final approval of the version to be published, and are accountable for all aspects of the work. MS and DS contributed equally and are joint last authors. DS and MS are the guarantors accept full responsibility for the work and/or the conduct of the study, had access to the data, and controlled the decision to publish.

**Funding** UPSIDES has received funding from the European Union's Horizon 2020 research and innovation programme under grant agreement No 779263.

**Competing interests** None declared.

**Patient and public involvement** Patients and/or the public were not involved in the design, or conduct, or reporting, or dissemination plans of this research.

**Patient consent for publication** Not applicable.

**Ethics approval** This study involves human participants and ethical approval was obtained by each site: Ulm University Ethics Commission (Application nr. 195/18), Mengo IRB Uganda (MH: 360; MH/REC/141/8/2018), National Institute for Medical Research Tanzania (NIMR/HQ/R.8a/Vol.IX/2982), Institutional Review Board, Ifakara Health Institute, Tanzania (IHI/IRB/No. 28-2018), Ärztekammer Hamburg, Germany (MC-230/18), Indian Council of Medical Research (Indo-foreign/66/M/2017-NCD-1), Indian Law Society (ILS/37/2018) and Human Subjects Research Committee of Ben-Gurion University (ref: 1621-2). Participants gave informed consent to participate in the study before taking part.

**Provenance and peer review** Not commissioned; externally peer reviewed.

**Data availability statement** Data are available on reasonable request. Full transcripts are not publicly available due to their containing information that could compromise the privacy of research participants.

**ORCID iDs**
Mary Ramesh http://orcid.org/0000-0002-4942-3923
Ashleigh Charles http://orcid.org/0000-0003-2222-4358
Alina Grayzman http://orcid.org/0000-0002-8192-7697
Ramona Hiltensperger http://orcid.org/0000-0003-2544-4188
Jasmine Kalha http://orcid.org/0000-0001-7357-2366
Arti Kulkarni http://orcid.org/0000-0002-3281-4388
Candelaria Mahlke http://orcid.org/0000-0001-9573-6106
Galia S Moran http://orcid.org/0000-0001-9718-1773
Richard Mpango http://orcid.org/0000-0001-6960-3174
Annabel S. Mueller-Stierlin http://orcid.org/0000-0003-2812-5115
Rebecca Nixdorf http://orcid.org/0000-0002-3064-8380
Grace Kathryn Ryan http://orcid.org/0000-0002-9310-3513
Donat Shamba http://orcid.org/0000-0001-7431-7199
Mike Slade http://orcid.org/0000-0001-7020-3434

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
