## [Reviewer comments · BMJ Open]

ARTICLE DETAILS

TITLE (PROVISIONAL)	Societal and organisational influences on implementation of mental health peer support work in low-income and high-income settings: a qualitative focus group study
AUTHORS	Ramesh, Mary; Charles, Ashleigh; Grayzman, Alina; Hiltensperger, Ramona; Kalha, Jasmine; Kulkarni, Arti; Mahlke, Candelaria; Moran, Galia; Mpango, Richard; Stierlin, Annabel; Nixdorf, Rebecca; Ryan, Grace; Shamba, Donat; Slade, Mike

VERSION 1 – REVIEW

REVIEWER	Kaiser, Bonnie Duke University, Duke Global Health Institute
REVIEW RETURNED	27-Dec-2021

GENERAL COMMENTS	This paper addresses the important topic of peer support workers and contributes perspectives from low-income settings, which are relatively under-represented in this literature. My main suggestions relate to strengthening the presentation of the results to give a clearer sense of the data: Throughout the results section, it would be helpful to give more of a sense of how shared (or not) the various themes are. To clarify, I'm not suggesting incorporating numeric reporting of findings since that can be misleading with qualitative studies. The current presentation of results has some themes stated in a just-so way and example quotations given (it is fairly quotation-heavy), which makes it difficult to tell to what extent the findings are broadly shared across sites and participants vs. there being one or two instances. For example, these are some of what I'm referring to as just-so statements of findings: "Sometimes the PSWs are rejected when they go to visit service users" and "Religious beliefs can also act as a barrier in implementing peer support work." It's difficult to tell whether these issues come up frequently across FGDs and across study sites. In some parts, it would also help to give more of a sense of where the findings are coming from and how they're grounded in the data by providing concrete examples or specifics that don't rely on quotations. For example, the section about resources (theme 2) has a couple quotations per section, but they're not necessarily powerful quotes per se (unlike in other sections where the quotes are really adding something). It might make more sense to instead put the word count towards describing in-text the range of specifics that came up in the data (e.g., in the first section that could indicate that resource shortages are barriers because of airtime, travel, and other specific issues that were mentioned).
--

	This is an example of what I consider a more of a nuanced description that clarifies themes outside of quotations: “These initiatives have enabled the PSWs to be known as role models in the community and have inspired hope to others. Additionally, the notion of knowledge from experience adds value to the potential contribution of the PSW and helps transform and enhance the value of lived experience.” Finally, it feels somewhat like the reader is given a laundry list of thematic findings within each theme, and they could be brought together more to provide a synthesis within each theme. The fact that each sub-theme is usually presented in one sentence followed by 1-3 quotes is likely what is creating this feeling of a list. Again, I think there doesn't need to be so much reliance on quotations, as there are other ways to demonstrate grounding in the data. And it would leave more words for analysis and synthesis to help bring together the overall story for the reader. It could also be helpful to include a figure/visual to bring together the findings. I don't personally find a codebook as useful to include compared to a table or figure that communicates the findings more. A separate concern is that reading through the results, it feels like the deductive themes are dominant over the inductive ones. By that, I mean there seem to be cross-cutting themes (like stigma in communities as well as among providers), but these were somewhat forced into separate categories based on the deductive framework that was the starting point. For example, in the staff section (theme 4), the results consist of one sentence in-text and two quotations, which just seems like it's not very data-driven of a decision to make it an entire theme. Similarly, the start of theme 3 sounds like a definition that's from the literature rather than the data. I don't know whether I would suggest reworking the results, but at least it would be good to include more description of how the deductive-inductive analysis methods were approached and bringing together the cross-cutting themes more in the discussion. I'd move the strengths and limitations paragraph to later in the discussion so that the crux of the discussion comes first. Please also add a short conclusion. Minor comments: Pg 4, line 36 has an incomplete sentence Pg 6, line 4: I'd remove where it says 29 total participants because it sounds like that's across sites, which is confusing. Pg 6, sampling: 2 FGDs per site isn't really the sampling strategy. That would be how you identified and selected the participants for the FGDs (i.e. purposively based on ____). To me, this section is an example of how the paper's structure and headings (which presumably are defined by the journal) aren't ideal for the study being described, since there's already a description of purposive sampling in the Participants section. Please address saturation, including within sites and not only for the whole sample. Results: my understanding is the conceptual framework was developed before the study was conducted, in which case it
--	--

	doesn't make sense to include in results but rather in methods. Alternatively, clarify if it was actually developed based on the study results. Somewhat unrelated, but I always think of conceptual frameworks as visual. This seems more like a set of principles or characteristics (which are important and interesting). Table 2: for FGDs, the n is the group. This is relevant to the title but also could be incorporated as far as the n for groups and the total # of participants within the table. Pg 8, line 43: I worry that the example of a nurse potentially being killed might reinforce stereotypes of dangerousness of mental health patients. Could more context perhaps be provided there? Pg 9, line 37: The name of the Kampala hospital is de-identified, but it's named on pg 5. I don't know how many PSWs there are and/or whether it's realistic to de-identify the hospital if it's the only site with PSWs, but in general I'd just try to be consistent there. There are a lot of mentions of the upcoming RCT throughout, which shouldn't be so much of a focus of this paper. Just mentioning it once for context is sufficient. This isn't related to this paper per se, but in general, you don't want to guarantee confidentiality can be maintained with FGD participants, since there are other participants present who might or might not maintain confidentiality the way the researchers will.
--	---

REVIEWER	DeHart, Dana University of South Carolina
REVIEW RETURNED	20-Jan-2022

GENERAL COMMENTS	BMJ Open Bmjopen-2021-058724 Societal and organizational influences on implementation of mental health peer support work in low-income and high-income settings: qualitative focus group study This multisite, international study examines influences on implementation of peer support for persons with severe mental illness in three high-income and two low-income sites (Hamburg, Ulm, Be'er Sheva, Dar es Salaam, Kampala). The study addresses an important topic using a novel sample. Study procedures were sound, with attention to issues such as trustworthiness of qualitative data. The depth in exploring findings, however, was somewhat disappointing. Only a few quotes were provided for each theme, and it was difficult to tell how cohesive or developed these themes were in the absence of more illustrative examples. The reader is left a bit unsure whether they have a solid understanding of these themes, and implications for the themes are not well explored in the Discussion section. This being said, the study does appear to have potential to contribute to the literature if the authors can draw these themes out better, and—in particular—go into greater detail regarding differences between high-income and low-income sites. Further, the authors should discuss more about how their themes replicate or add new information to the literature and implications this has for research, practice, and policy. While page limitations may be a consideration, some of the appendices (and possibly even some of the tables) could be eliminated or abbreviated. In particular,
---

	Table 3, which lists the codebook, includes a lot of codes that are not discussed in the manuscript and could be reduced to just those codes pertinent to this write-up. Table 1 is helpful, but not essential if a more cursory explanation could be integrated into the narrative, and the appendices seem unnecessary. There are a few places where the numbers for the sample don't seem to add up (in the Abstract and Table 2). Limitations should mention the need to conduct studies of persons supported, not just professionals and peer supports.
--	--

REVIEWER	Easton, Katherine University of Sheffield, School of Education
REVIEW RETURNED	27-Jan-2022

GENERAL COMMENTS	The authors report on a study to identify societal and organisational influences on PHPW in low and high income countries. The research focus is novel and sampling is comprehensive. The paper has many strengths. I particularly liked the style of writing which I found to be accessible, clear and concise. Often authors can use jargon and overly complicate topics but this paper avoids that. I do think that the analysis of the data could be developed a little further to identify higher level codes and move away for initial superficial coding. For example, it wasn't clear to me from the reporting how the authors felt the findings linked back to the low/high income aspect of the research. Since this was a particular defining element of the research I thought that it might come through more in the results. The results feel a little brief considering there must have been a huge amount of data collected. I'm not sure if this is the result of an imposed word limit or the limited introduction to each theme/quote with no links to literature/theory. It might have been better to focus on 2 main themes in the paper and expand on them and then have a supplementary document with additional themes and data. I'm not suggesting new analysis however, I feel the reader will benefit more from the paper if the authors can elaborate on the main themes and how the findings link to the low/high income element of the research. With that said, the paper was a pleasure to read. The background and methods will be of value to students in this area and the focus will inspire additional research.
--

VERSION 1 – AUTHOR RESPONSE

Reviewer: 1 Dr. Bonnie Kaiser, Duke University Comments to the Author: This paper addresses the important topic of peer support workers and contributes perspectives from low-income settings, which are relatively underrepresented in this literature.		
1	My main suggestions relate to strengthening the presentation of the results to give a clearer sense of the data: Throughout the results section, it would be helpful to give more of a sense of how shared (or not) the various themes are. To clarify, I'm not suggesting incorporating numeric reporting of findings since that can be misleading with qualitative studies. The current presentation of results has some themes stated in a just-	This has been revised, more description is provided on the manuscript
	so way and example quotations given (it is fairly quotationheavy), which makes it difficult to tell to what extent the findings are broadly shared across sites and participants vs. there being one or two instances. For example, these are some of what I'm referring to as just-so statements of findings: "Sometimes the PSWs are rejected when they go to visit service users" and "Religious beliefs can also act as a barrier in implementing peer support work." It's difficult to tell whether these issues come up frequently across FGDs and across study sites.	

2	In some parts, it would also help to give more of a sense of where the findings are coming from and how they're grounded in the data by providing concrete examples or specifics that don't rely on quotations. For example, the section about resources (theme 2) has a couple quotations per section, but they're not necessarily powerful quotes per se (unlike in other sections where the quotes are really adding something). It might make more sense to instead put the word count towards describing in text the range of specifics that came up in the data (e.g., in the first section that could indicate that resource shortages are barriers because of airtime, travel, and other specific issues that were mentioned). This is an example of what I consider a more of a nuanced description that clarifies themes outside of quotations: "These initiatives have enabled the PSWs to be known as role models in the community and have inspired hope to others. Additionally, the notion of knowledge from experience adds value to the potential contribution of the PSW and helps transform and enhance the value of lived experience."	This has been addressed
3	Finally, it feels somewhat like the reader is given a laundry list of thematic findings within each theme, and they could be brought together more to provide a synthesis within each theme. The fact that each sub-theme is usually presented in one sentence followed by 1-3 quotes is likely what is creating this feeling of a list. Again, I think there doesn't need to be so much reliance on	Thank you, we have added more quotes for each themes, and more description of the findings.
	quotations, as there are other ways to demonstrate grounding in the data. And it would leave more words for analysis and synthesis to help bring together the overall story for the reader. It could also be helpful to include a figure/visual to bring together the findings. I don't personally find a codebook as useful to include compared to a table or figure that communicates the findings more.	

4	A separate concern is that reading through the results, it feels like the deductive themes are dominant over the inductive ones. By that, I mean there seem to be cross-cutting themes (like stigma in communities as well as among providers), but these were somewhat forced into separate categories based on the deductive framework that was the starting point. For example, in the staff section (theme 4), the results consist of one sentence in text and two quotations, which just seems like it's not very data-driven or a decision to make it an entire theme. Similarly, the start of theme 3 sounds like a definition that's from the literature rather than the data. I don't know whether I would suggest reworking the results, but at least it would be good to include more description of how the deductive-inductive analysis methods were approached and bringing together the crosscutting themes more in the discussion.	Thank you, Stigma & community attitude and stigma from – community is combined
5	I'd move the strengths and limitations paragraph to later in the discussion so that the crux of the discussion comes first. Please also add a short conclusion.	Strengths and limitations has been moved and is now after the discussion and conclusion section
6	Minor comments: Pg 4, line 36 has an incomplete sentence	Thank you, the sentence is completed. Lower income and higher income settings.
7	Pg 6, line 4: I'd remove where it says 29 total participants because it sounds like that's across sites, which is confusing.	This has been revised
8	Pg 6, sampling: 2 FGDs per site isn't really the sampling strategy. That would be how you identified and selected the participants for the FGDs (i.e. purposively based on ____). To me, this section is	Sampling section has been deleted
	an example of how the paper's structure and headings (which presumably are defined by the journal) aren't ideal for the study being described, since there's already a description of purposive sampling in the Participants section.	

9	Please address saturation, including within sites and not only for the whole sample.	Thank you, this has been added a study limitation. Two FGDs per site may not reach saturation. However, the study involved different set of respondents to represent different groups who either had an experience in peer support workers or planning to use peer support worker.
10	Results: my understanding is the conceptual framework was developed before the study was conducted, in which case it doesn't make sense to include in results but rather in methods. Alternatively, clarify if it was actually developed based on the study results. Somewhat unrelated, but I always think of conceptual frameworks as visual. This seems more like a set of principles or characteristics (which are important and interesting).	Thank you, We have deleted details of the conceptual framework on the results section.
11	Table 2: for FGDs, the n is the group. This is relevant to the title but also could be incorporated as far as the n for groups and the total # of participants within the table.	Thank you, Revisions have been made on the table for FGD participants characteristics
12	Pg 8, line 43: I worry that the example of a nurse potentially being killed might reinforce stereotypes of dangerousness of mental health patients. Could more context perhaps be provided there?	Thank you for this note, this part of the quote is deleted
13	Pg 9, line 37: The name of the Kampala hospital is de-identified, but it's named on pg 5. I don't know how many PSWs there are and/or whether it's realistic to de-identify the hospital if it's the only site with PSWs, but in general I'd just try to be consistent there.	We have revised this on the participants subheading in page 5.
14	There are a lot of mentions of the upcoming RCT throughout, which shouldn't be so much of a focus of this paper. Just mentioning it once for context is sufficient.	Details of the coming UPSIDES RCT have been deleted
15	This isn't related to this paper per se, but in general, you don't want to guarantee confidentiality can be maintained with FGD participants, since there are other participants present who might or might not maintain confidentiality the way the researchers will.	Thank you, this is noted

Reviewer: 2 Dr. Dana DeHart, University of South Carolina	Comments to the Author: BMJ Open Bmjopen-2021-058724 Societal and organizational influences on implementation of mental health peer support work in low-income and high-income settings: qualitative focus group study	
1	This multisite, international study examines influences on implementation of peer support for persons with severe mental illness in three high-income and two low-income sites (Hamburg, Ulm, Be'er Sheva, Dar es Salaam, Kampala). The study addresses an important topic using a novel sample. Study procedures were sound, with attention to issues such as trustworthiness of qualitative data. The depth in exploring findings, however, was somewhat disappointing. Only a few quotes were provided for each theme, and it was difficult to tell how cohesive or developed these themes were in the absence of more illustrative examples. The reader is left a bit unsure whether they have a solid understanding of these themes, and implications for the themes are not well explored in the Discussion section. This being said, the study does appear to have potential to contribute to the literature if the authors can draw these themes out better, and— in particular—go into greater detail regarding differences between high-income and low-income sites.	Thank you, we have added more quotes for each themes, and more description of the findings.

2	Further, the authors should discuss more about how their themes replicate or add new information to the literature and implications this has for research, practice, and policy. While page limitations may be a consideration, some of the appendices (and possibly even some of the tables) could be eliminated or abbreviated. In particular, Table 3, which lists the codebook, includes a lot of codes that are not discussed in the manuscript and could be reduced to just those codes pertinent to this writeup. Table 1 is helpful, but not essential if a more cursory explanation could be integrated into the narrative, and the appendices seem unnecessary.	Thank you, table 1 is removed – details have been added in the narratives
----------	---	---

3	There are a few places where the numbers for the sample don't seem to add up (in the Abstract and Table 2). Limitations should mention the need to conduct studies of persons supported, not just professionals and peer supports.	We have revised the sample size in the abstract and Table for the characteristics of focus group participants. We have also revised the study limitations to include those who use peer support services.
Reviewer: 3 Dr. Katherine Easton, University of Sheffield		
1	Comments to the Author: The authors report on a study to identify societal and organisational influences on PHPW in low and high income countries. The research focus is novel and sampling is comprehensive. The paper has many strengths. I particularly liked the style of writing which I found to be accessible, clear and concise. Often authors can use jargon and overly complicate topics but this paper avoids that.	Thank you,
2	I do think that the analysis of the data could be developed a little further to identify higher level codes and move away for initial superficial coding. For example, it wasn't clear to me from the reporting how the authors felt the findings linked back to the low/high income aspect of the research. Since this was a	Thank you, we have added more information on the themes and comparing results between lower income and higher income settings.
	particular defining element of the research I thought that it might come through more in the results. The results feel a little brief considering there must have been a huge amount of data collected. I'm not sure if this is the result of an imposed word limit or the limited introduction to each theme/quote with no links to literature/theory. It might have been better to focus on 2 main themes in the paper and expand on them and then have a supplementary document with additional themes and data. I'm not suggesting new analysis however, I feel the reader will benefit more from the paper if the authors can elaborate on the main themes and how the findings link to the low/high income element of the research.	
3	With that said, the paper was a pleasure to read. The background and methods will be of value to students in this area and the focus will inspire additional research.	Thank you. We are grateful for this positive feedback.

VERSION 2 – REVIEW

REVIEWER	Kaiser, Bonnie Duke University, Duke Global Health Institute
REVIEW RETURNED	23-May-2022

GENERAL COMMENTS	While some revisions have been made, they are relatively limited in scope, and my substantive comments from the first round of review still stand. Specifically, comments #1-5 in your table haven't been addressed. Minor issues: Please update the Abstract and Results so that the n is the number of focus groups rather than individuals. It's fine to include also the breakdown of individuals by role/category, but the primary n (presented first) should be the group. In addition to mentioning that saturation might not have been reached, could you assess saturation? Are new themes arising even as you analyze the last FGD or two, or did you reach a point of only repetition of themes? Theme 1 seems to be renamed to include staff but not really reworked to include staff as far as the results reporting.
--

REVIEWER	Easton, Katherine University of Sheffield, School of Education
REVIEW RETURNED	30-May-2022

GENERAL COMMENTS	Thank you for resubmitting your manuscript. There have been some minor changes in the presentation of the finding. These read more clearly now and you have added some detail on the relationship between low and high income settings. As noted previously, the paper is written in an easy to digest manner and will make for a nice reference point for students studying in this area with respect to methods and implications.
---

VERSION 2 – AUTHOR RESPONSE

Reviewer: 1 Dr. Bonnie Kaiser, Duke University		
--	--	--

1	Comments to the Author: While some revisions have been made, they are relatively limited in scope, and my substantive comments from the first round of review still stand. Specifically, comments #1-5 in your table haven't been addressed	The suggested revisions from the first round of review have been made. We have addressed the comments that were given in the first round of review and revisions have been made in the manuscript including adding more details to the findings.
2	Minor issues: Please update the Abstract and Results so that the n is the number of focus groups rather than individuals. It's fine to include also the breakdown of individuals by role/category, but the primary n (presented first) should be the group.	Thank you. Revisions have been made as suggested.
3	In addition to mentioning that saturation might not have been reached, could you assess saturation? Are new themes arising even as you analyze the last FGD or two, or did you reach a point of only repetition of themes?	Saturation may not have been reached as per the sample size of the study; we acknowledge this as a limitation of the study.
4	Theme 1 seems to be renamed to include staff but not really reworked to include staff as far as the results reporting.	We have reworked on this theme and have included staff attitudes in the reporting of results.
Reviewer: 3 Dr. Katherine Easton, University of Sheffield		
1	Comments to the Author: Thank you for resubmitting your manuscript. There have been some minor changes in the presentation of the finding. These read more clearly now and you have added some detail on the	Thank you. We are grateful for the positive feedback.

	relationship between low- and high-income settings.	
	As noted previously, the paper is written in an easy to digest manner and will make for a nice reference point for students studying in this area with respect to methods and implications.	

VERSION 3 – REVIEW

REVIEWER	Kaiser, Bonnie Duke University, Duke Global Health Institute
REVIEW RETURNED	06-Nov-2022

GENERAL COMMENTS	There have been additional improvements. I particularly appreciate the added section on staff attitudes under theme 1. I would slightly rearrange that section so that the negatives and positives regarding community are together, then the same (negatives and positives) for the section on staff. Otherwise it feels a little like it's jumping around community-staff-community-staff. I'd really like to see the types of changes that were incorporated here-and-there expanded throughout the results. Some sections are reading much better now, but many sections remain unchanged. For example, both reviewer 2 and I asked for more synthesis (rather than a brief list of sub-themes with accompanying quotations) and more analysis. We're really looking for the authors to do the work of synthesizing the data and communicating to readers what constitutes those themes beyond a sentence and a quotation. There is variability in how well that was changed. For example, while the first theme is stronger, the Organisational Culture and Training and Support themes still read like a laundry list of issues with accompanying quotes, without the analysis and synthesis to tie those issues together. I feel like as a reader, I'm having to do extra work to try to figure out how these things are linked and what the complications are, etc. I don't think all of the quotations are that strong, so I would be more strategic about which ones to include and putting more text into synthesis and tying together the theme in a more coherent way. Also, under theme 2, there is a sentence summary: "Participants from higher income countries also reported that, whilst PSWs are an important component of mental health services, there is a limited budget set for them and there is a particular challenge in relation to funding arrangements for the peer support program." The accompanying quotations suggest there is a lot going on there, and I'd like to hear the authors summarize these sub-
---

themes (whether with or without those quotations). I'd love more synthesis of what's in the overall data here (rather than only what's in these selected quotations). Is it that PSWs sometimes aren't paid, that they can't fulfill all the roles this program is designed to entail, that there are refusals to provide the funding, does this result in breaks in care provision, etc? I understand that the overall sub-theme is financial limitations, but I have a limited sense of what that actually means, and even a couple more sentences could answer that.

There is still a need for consistent communication of the commonality and coherence of themes (note, this does not just mean whether the same themes arose in both high- and low-income country settings). This has been partially addressed, though the response suggests this was addressed by adding more quotations. The central request is to communicate to readers how shared (vs rare) particular experiences or themes are. This was certainly improved, for example with additions like "Some participants especially in lower income countries, reported that." However, with the reporting of other results, it can give the impression that either everyone said something or that an issue was only raised in that one quotation (and it's hard to tell which is which), whereas a bit of editing could communicate what is in the data beyond that quotation. For example, theme 4 (role definition) includes two statements with accompanying quotations from the same site. Did it come up elsewhere? You don't need quotations to reflect that; you can just give a sense of how common or rare themes are. Similarly, is the quote about religion in theme 1 the one time that came up? Same question about positive community initiatives? Or were these themes raised in all 6 FGDs in low-income countries?

I'm not looking for a number but just a sense of whether this was reported often vs. once vs. a few times. The staff section is a nice example: in the first half (negative staff attitudes), it's unclear but seems to suggest all staff are stigmatizing. Then later, it says "Despite the fact that there are some health providers who label peer support workers, there are also others who think that peer support workers are an asset to both health service providers and the recipients of peer support services. One participant from a lower income country perceived that..." This latter description is more nuanced and reflective of diversity within the data. I encourage the authors to focus on such instances of communicating findings beyond relying on quotations, as it does a lot to help readers get a fuller sense of the data. I want to emphasize that this in no way requires a whole rewrite but just some very minor additions here and there in the text.

The issue of reporting sample sizes incorrectly still needs to be addressed: with focus group discussions, the n is the group, not the number of individuals. This reflects that within each sampling unit (group), the data collected are influenced by group dynamics and the presence of other people, so it's not equivalent to a study involving 86 interviews. Including a breakdown as in Table 2 is fine, but in the abstract and in-text where the sample size is mentioned, it should be the number of FGDs rather than individuals who participated.

The figure isn't really adding anything, as it's just a list of themes. If they can be linked in ways that communicate the meaningful

	connections among themes, that would be great. If not, I would just cut it.
--	---

VERSION 3 – AUTHOR RESPONSE

Reviewer: 1

Dr. Bonnie Kaiser, Duke University

1 There have been additional improvements. I particularly appreciate the added section on staff attitudes under theme 1. I would slightly rearrange that section so that the negatives and positives regarding community are together, then the same (negatives and positives) for the section on staff. Otherwise it feels a little like it's jumping around community-staff-community-staff. We thank the reviewer for their suggestion, and we have now rearranged that section so that the negatives and positives regarding community and staff are together: page 7,8.

2 I'd really like to see the types of changes that were incorporated here-and-there expanded throughout the results. Some sections are reading much better now, but many sections remain unchanged. For example, both reviewer 2 and I asked for more synthesis (rather than a brief list of sub-themes with accompanying quotations) and more analysis. We're really looking for the authors to do the work of synthesizing the data and communicating to readers what constitutes those themes beyond a sentence and a quotation. There is variability in how well that was changed. For example, while the first theme is stronger, the Organisational Culture and Training and Support themes still read like a laundry list of issues with accompanying quotes, without the analysis and synthesis to tie those issues together. I feel like as a reader, I'm having to do extra work to try to figure out how these things are linked and what the complications are, etc. I don't think all of the quotations are that strong, so I would be more strategic about which ones to include and putting more text into synthesis and tying together the theme in a more coherent way. We thank the reviewer for their suggestions and 'Organizational Culture' has now been synthesized further to tie issues together: page, 10, 11.

'Training and Support' has now been synthesized to tie issues together: page 12, 13.

3 Also, under theme 2, there is a sentence summary: "Participants from higher income countries also reported that, whilst PSWs are an important component of mental health services, there is a limited budget set for them and there is a particular challenge in relation to funding arrangements for the peer support program." The accompanying quotations suggest there is a lot going on there, and I'd like to hear the authors summarize these sub-themes (whether with or without those quotations). I'd love more synthesis of what's in the overall data here (rather than only what's in these selected quotations). Is it that PSWs sometimes aren't paid, that they can't fulfill all the roles this program is designed to entail, that there are refusals to provide the funding, does this result in breaks in care provision, etc? I understand that the overall sub-theme is financial limitations, but I have a limited sense of what that actually means, and even a couple more sentences could answer that. We have now added further synthesis to help the reader further understand the sub-theme: page 10.

4 There is still a need for consistent communication of the commonality and coherence of themes (note, this does not just mean whether the same themes arose in both high- and low-income country settings). This has been partially addressed, though the response suggests this was addressed by adding more quotations. The central request is to communicate to readers how shared (vs rare) particular experiences or themes are. This was certainly improved, for example with additions like "Some participants especially in lower income countries, reported that." However, with the reporting of other results, it can give the impression that either everyone said something or that an issue was only raised in that one quotation (and it's hard to tell which is which), whereas a bit of editing could communicate what is in the data beyond that quotation. For example, theme 4 (role definition) includes two statements with accompanying quotations from the same site. Did it come up elsewhere? You don't need quotations to reflect that; you can just give a sense of how common or

rare themes are. Similarly, is the quote about religion in theme 1 the one time that came up? Same question about positive community initiatives? Or were these themes raised in all 6 FGDs in low-income countries? For theme 4 (role definition) we have added further details if this came up elsewhere: page 12.

For theme 1 (Community and staff attitudes) we have added further details in relation to religious beliefs: page 7.

For theme 1 (Community and staff attitudes) we have added further details: page 7.

For theme 1 (Community and staff attitudes) we have amended the wording around religious beliefs: page 7, paragraph 6.

We have added two quotes from Dar es Salaam to improve the consistency and coherence of themes: page 7, paragraph 7, and page 10, paragraph 4.

I'm not looking for a number but just a sense of whether this was reported often vs. once vs. a few times. The staff section is a nice example: in the first half (negative staff attitudes), it's unclear but seems to suggest all staff are stigmatizing. Then later, it says "Despite the fact that there are some health providers who label peer support workers, there are also others who think that peer support workers are an asset to both health service providers and the recipients of peer support services. One participant from a lower income country perceived that..." This latter description is more nuanced and reflective of diversity within the data. I encourage the authors to focus on such instances of communicating findings beyond relying on quotations, as it does a lot to help readers get a fuller sense of the data. I want to emphasize that this in no way requires a whole rewrite but just some very minor additions here and there in the text. We thank the reviewer for their comments and have now incorporated this throughout the results section: page 7, 8, 9, 10, 11, 12, 13.

The issue of reporting sample sizes incorrectly still needs to be addressed: with focus group discussions, the n is the group, not the number of individuals. This reflects that within each sampling unit (group), the data collected are influenced by group dynamics and the presence of other people, so it's not equivalent to a study involving 86 interviews. Including a breakdown as in Table 2 is fine, but in the abstract and in-text where the sample size is mentioned, it should be the number of FGDs rather than individuals who participated. We have now addressed the sample size: page 2 (abstract, participants), 3 (strengths and limitations), 7 (introduction to results).

The figure isn't really adding anything, as it's just a list of themes. If they can be linked in ways that communicate the meaningful connections among themes, that would be great. If not, I would just cut it. We thank the reviewer and have now decided to not include the figure.

We have made several changes to sentences/wording throughout the manuscript.

We have used peer support workers (PSWs) throughout the manuscript for consistency: page 1.

We changed the types of setting to tertiary and secondary mental health care: page 1.

We have deleted the Indian Ethics Board in ethical approval as participants from India were not included in the focus groups for this paper: page 6, paragraph 3.